# *Tillandsia usneoides* Extract Decreases the Primary Tumor in a Murine Breast Cancer Model but Not in Melanoma

**DOI:** 10.3390/cancers14215383

**Published:** 2022-11-01

**Authors:** Paola Lasso, Laura Rojas, Cindy Arévalo, Claudia Urueña, Natalia Murillo, Alfonso Barreto, Geison M. Costa, Susana Fiorentino

**Affiliations:** 1Grupo de Inmunobiología y Biología Celular, Pontificia Universidad Javeriana, Bogotá 110231, Colombia; 2Grupo de Investigación en Fitoquímica, Pontificia Universidad Javeriana, Bogotá 110231, Colombia

**Keywords:** breast cancer, melanoma, *Tillandsia usneoides*, immunomodulation, antitumor, plant extracts

## Abstract

**Simple Summary:**

Cancer is a major public health problem worldwide and one of the major causes of mortality. Current therapies are becoming ineffective due to intratumoral heterogeneity, drug resistance, toxic effects, and relapses. Therefore, new therapeutic alternatives or their combination with conventional therapy are being explored. *T. usneoides* extract regulates the metabolism of the 4T1 and B16-F10 cell lines in an antagonistic manner with a significant impact on the tumor microenvironment, apparently related to the enhancement of an effective antitumor immune response, that allow the reduction of the 4T1 tumor but not of B16-F10. These data not only allow scientific validation of traditional knowledge but also take advantage of it to continue discovering mixtures of metabolites with antitumor and immunomodulatory activity.

**Abstract:**

The main limits of current antitumor therapies are chemoresistance, relapses, and toxicity that impair patient quality of life. Therefore, the discovery of therapeutic alternatives, such as adjuvants to conventional therapy that modulate the intracellular oxidation state or the immune response, remains a challenge. Owing to traditional medicine, several uses of plants are known, indicating a promising antitumor and immunomodulatory effect. We evaluated the effect of ethanolic extract of *T. usneoides* in vitro and in vivo in models of 4T1 breast cancer and B16-F10 melanoma. In vitro evaluations with both cell lines showed that the extract has cytotoxic activity and induces apoptotic cell death. However, its effect on ROS production and glucose uptake was opposite. In vivo, only in the 4T1 model, a significant decrease in tumor size was found in animals treated with the extract, accompanied by an increase in dendritic cells and activated CD8^+^ T cells, and a decrease in myeloid-derived suppressor-like cells (MDSC-LC) and Tregs in the tumor microenvironment. These results suggest that *T. usneoides* extract antagonistically regulates tumor metabolism of 4T1 vs. B16-F10, impacting the tumor microenvironment and effective antitumor immune response, leading to a reduction in 4T1 tumor size but not on B16-F10.

## 1. Introduction

Cancer is one of the major causes of mortality, accounting for nearly 10 million deaths in 2020 or nearly one in six deaths worldwide [1]. Additionally, the reports show that by 2040 there will be 27.5 million new cancer patients each year [2]. Current therapies are becoming ineffective due to intratumoral heterogeneity, drug resistance, toxic effects, and relapses. In addition to the treatments already known for the treatment of cancer, some plants stand out in traditional medicine for their potential antitumor effect. The natural products obtained from plants have been a source of medicines for many years, and more recently, their use in complex mixtures that allow their activity to be enhanced has been proposed [3,4,5].

*Tillandsia usneoides* is a Bromeliaceae, commonly known as Spanish moss, that is distributed in Central and northern South American populations including Colombia [6,7]. It has traditionally been used in the control of diabetes [8] and its activity is attributed to the presence of 3-hydroxy-3-methylglutaric acid (HMG), a highly toxic compound [9]. However, other compounds with hypoglycemic activity, possibly exerted through GLUT4 modulation and with low cellular toxicity, have been identified, such as 5,7,4′-trihydroxy-3,6,3′,5′-tetramethoxyflavone (Flav1) [10]. Although antitumor effects for *Tillandsia recurvata* were reported in 2007 [11], not much progress has been made in understanding the mechanisms involved in this activity.

The antitumor activity of plant-derived extracts and compounds has sometimes been attributed, in a very simple way, to their antioxidant capacity. However, recent omics approaches have revealed multiple biological activities related to cancer control, among which the ability to modulate the immune response and the tumor microenvironment stands out [3,12]. Tumors can escape from the immune response by inhibiting T cell expansion and function, finally leading to immunosuppression. Myeloid-derived suppressor cells (MDSCs) are a subset of immature myeloid cells with potent immunosuppressive capacity present at different levels in all cancer types [13]. These cells lead to an immunosuppressive tumor microenvironment (TME) with attenuation of the antitumor immune response, especially of T cells, thus promoting cancer progression. In patients with a wide variety of malignancies, MDSCs have been reported to correlate with disease progression and negatively impact overall survival. Therefore, MDSCs have become an important therapeutic target [14].

It has been reported that triterpenes and polyphenols enhance effector cell-mediated immune response, antigen presentation, and T cell recognition [15,16,17]. These compounds have been identified as the main components of *T. usneoides* ethanolic extract, suggesting that this extract may have an important immunomodulatory activity. Considering the wide distribution of *T. usneoides* in our country, Columbia, we wanted to evaluate its antitumor and immunomodulatory activity in two murine tumor models. Surprisingly, we found differential activity of this plant extract, confirming the fact that complex mixtures can be specific, supporting the continued development of polymolecular drugs from crude plant extracts.

## 2. Materials and Methods

### 2.1. Plant Material

Fresh leaves of *Tillandsia usneoides* were collected in Villa de Leyva, Boyacá, Colombia, and identified by the Javeriana University Herbarium (voucher specimen number 30547). The extract of *T. usneoides* was obtained and then chemically characterized. The P2Et was produced and characterized as prescribed from *Caesalpinia spinosa* [18,19]. Fresh pods of *C. spinosa* were collected in Villa de Leyva, Boyacá, Colombia, and identified by the Colombian National Herbarium (voucher specimen number COL 588448, Contract for Access to Genetic Resources and Derivate Products for Scientific Research without Commercial Interest, number 220 of 2018).

### 2.2. Ultra-Performance Liquid Chromatography–Photodiode Array Detection (UPLC–PDA) Conditions

The UPLC–PDA analysis was performed with an Acquity UPLC H-class (Waters, Milford, MA, USA) equipped with an photodiode array detector, quaternary pump, on-line degasser, and autosampler. Chromatographic separation was performed on a Phenomenex^®^ Kinetex C18 column (100 × 2.1 mm, 1.7 µm) at 25 ± 1 °C with a linear gradient elution of acetonitrile (Solvent A) and 0.1% formic acid in water (Solvent B), as follows: 0 to 20 min, 17 to 70% A; 20 to 30 min, 70 to 95% A; 30 to 32 min, 95% A; and 32 to 40 min, 95 to 17% A. The volume of injection was 3 µL and the flow rate was 0.4  mL/min. The wavelength used was 274 nm, 305 nm, and 350 nm, with spectra acquired over a range of 200–450 nm.

### 2.3. In Vitro Cytotoxicity Assays

The cytotoxic effect of *T. usneoides* extract on tumor cells was evaluated using methylthiazol tetrazolium (MTT) assay (Sigma-Aldrich, Saint Louis, MO, USA) as previously reported [20]. The IC_50_ value (50% inhibition of cell growth) was calculated using GraphPad Prism version 8.1.1 for Mac OS X statistics software (GraphPad Software, San Diego, CA, USA).

### 2.4. Annexin V and PI Double-Staining Assay

Phosphatidylserine (PS) externalization was assessed by flow cytometry using Annexin V (Molecular Probes, Invitrogen Corp, Carlsbad, CA, USA) and propidium iodide (PI) (Sigma, Saint Louis, MO, USA) as previously reported [19]. Briefly, 2 × 10^5^ cells were treated with the IC_50_ and IC_50_/2 of the *T. usneoides* extract, doxorubicin (positive control, 0.4748 µM for 4T1 and 0.044 µM for B16-F10), and DMSO or ethanol (negative controls, 0.02%), for 24 h. After treatment, the cells were resuspended in annexin buffer (100 mM Hepes, 140 mM NaCl, 2.5 mM CaCl_2_) and incubated with annexin V-FITC for 8 min at room temperature. Then, cells were incubated with PI for 3 min at 4 °C. Finally, the samples were acquired on a FACSAria II-U (BD) flow cytometer and analyzed with the FlowJo v10.8.1 software (BD Life Sciences, Franklin Lakes, NJ, USA). The assays were performed in triplicate.

### 2.5. ROS Measurement

To evaluate ROS production, 2 × 10^5^ cells were plated in 6-well plates and treated with the IC_50_ and IC_50_/5 of the *T. usneoides* extract, IC_50_ (34.1 µg/mL for 4T1 and 36.4 µg/mL for B16-F10) and IC_50_/5 of the P2Et extract (anti-oxidant control), doxorubicin (pro-oxidant control, 0.2374 µM for 4T1 and 22 nM for B16-F10), and DMSO or ethanol (negative controls, 0.02%) for 6 h, 12 h, and 24 h. Cells were stained with 1 µM 2′,7′ diclorodihidrofluoresceina diacetato (H_2_DCFDA) (Sigma Aldrich, Saint Louis MO, USA) for 40 min at 37 °C, followed by PI (Sigma-Aldrich). Each sample was then acquired using an FACSAria II-U (BD) and analyzed with FlowJo v10.8.1 software (BD Life Sciences). Experiments were performed in triplicate on three independent experiments and the results were expressed as mean ± SEM.

### 2.6. Glucose Uptake Assay

For evaluation of glucose uptake, 1 × 10^6^ cells were seeded on 12-well plates and incubated for 6 h and 12 h with the IC_50_ and IC_50_/5 of the *T. usneoides* extract, IC_50_ (34.1 µg/mL for 4T1 and 36.4 µg/mL for B16-F10) and IC_50_/5 of the P2Et extract (positive control), rotenone (positive control, 1 µM for 4T1 and 50 µM for B16-F10), and DMSO or ethanol (negative controls, 0.02%). After treatments, cells were removed by trypsinization, PBS washed, and resuspended in 40 µM of 2-NBDG [2-(N-(7-Nitrobenz-2-oxa-1,3-diazol-4-il) amino)-2-desoxiglucosa] (Invitrogen Molecular Probes) prepared in RPMI 1640 without phenol red. Then, cells were incubated for 30 min at 37 °C and washed with cold PBS 1X. Live versus dead cell discrimination labeling was performed with PI (Sigma-Aldrich). Immediately, samples were acquired by FACSAria II-U (BD) flow cytometer and analyzed with FlowJo v10.8.1 software (BD Life Sciences). Experiments were performed in triplicate on two independent experiments and the results were expressed as mean ± SEM.

### 2.7. Measurement of Mitochondrial Membrane Potential

Mitochondrial membrane potential (MMP) was measured in 4T1 and B16-F10 cell lines by flow cytometry using JC-1 dye (Sigma, St. Louis, MO, USA) as previously reported [20]. Briefly, 1 × 10^5^ cells were treated with the IC_50_ and IC_50_/5 of the *T. usneoides* extract, IC_50_ (34.1 µg/mL for 4T1 and 36.4 µg/mL for B16-F10) and IC_50_/5 of the P2Et extract (positive control), valinomycin (positive control, 1 μg/mL), and DMSO or ethanol (negative controls, 0.02%) for 6 h and 12 h. JC-1 (2.5 μg/mL in PBS) was added and incubated for 10 min at 37 °C. The cells were acquired on a FACSAria II-U (Becton Dickinson, BD, Franklin Lakes, NJ, USA) and analyzed with FlowJo v10.8.1 software (BD Life Sciences), which calculated the red/green fluorescence ratios. JC-1 aggregates were evaluated in FL-2 (585 nm), showing a normal mitochondrial membrane potential, while an increase in FL-1 (530 nm) fluorescence was associated with monomers due to loss of the mitochondrial membrane potential. Experiments were performed in triplicate and the results were expressed as mean ± SEM.

### 2.8. Mice

Female C57BL/6NCrl and BALB/cAnNCrl young (10 to 12 weeks old) mice were housed at the animal facilities of the Pontificia Universidad Javeriana (PUJ, Bogotá, Colombia) following the established protocols of the Ethics Committee of the Faculty of Sciences and National and International Legislation for Live Animal Experimentation (Colombia Republic, Resolution 08430, 1993; National Academy of Sciences, 2010). Each protocol was approved by the animal experimentation committee of PUJ (FUA-093-20). 

### 2.9. Tumor Cell Lines and Culture Conditions

4T1 and B16-F10 cells were cultured in RPMI-1640 medium (Eurobio, Toulouse, France) with 10% heat-inactivated fetal bovine serum (FBS), 2 mM L-glutamine, 100 U/mL penicillin, 100 μg/mL streptomycin, 0.01 M HEPES buffer, and 1 mM sodium pyruvate (Eurobio) and cultivated in a humidified incubator at 37 °C in 5% CO_2_.

### 2.10. Abs

The Abs used for cell-surface staining included: anti-CD3 Pacific Blue (clone 17A2), anti-CD8 PE Dazzle 594 (clone 53.6.7), anti-CD45 PE-Cy5 (clone 30-F11), anti-Ly-6G PE-Cy7 (clone 1A8), anti-Ly-6C APC-Cy7 (clone AL-21), anti-PD-L1 PE (clone 10F.9G2), anti-PD-1 APC (clone 29F-1A12), CD11b Alexa Fluor 700 (clone M1/70), anti-CD4 Brilliant Violet 570 (clone RM4-5), anti-CD44 PE-Cy7 (clone IM7), CD25 APC (clone 3C7) (Biolegend, San Diego, CA, USA), and CD11c FITC (clone HL3) (BD Biosciences, San José, CA, USA). The abs for intracellular staining were anti-FoxP3 Alexa Fluor 488 (clone MF23) (BD Biosciences) and anti-CTLA-4 PE (clone UC10-4F10-11) (Biolegend). A LIVE/DEAD Fixable Aqua Dead Cell Stain Kit (Life Technologies, Thermo Scientific, Eugene, OR, USA) was used for dead cell exclusion. The abs used for intracellular cytokines evaluation were anti-IFNγ Alexa Fluor 700 (clone XMG1.2), TNFα PE-Cy7 (clone MP6-XT22), IL-2 FITC (clone JES6-5H4) (BD Biosciences), anti-perforin (clone S16009A), and anti-granzyme B (QA16A02) (Biolegend).

### 2.11. Acute Toxicity Evaluation

Female BALB/cAnNCrl and C57BL/6NCrl mice (6 to 12 weeks of age) were divided into groups and intraperitoneally (IP) inoculated with 2000 mg/kg of *T. usneoides* extract. Lethal dose 50% (LD_50_) was calculated with Probit version 14 (Minitab Inc.). To ensure no toxicity, animals were treated with 142.5 mg/Kg 4T1 body weight of *T. usneoides* extract which corresponds to 4 times lower doses than lethal dose-50 (LD_50_). The dose of P2Et extract was used as previously reported [21].

### 2.12. In Vivo Tumor Development Experiments and Treatment

For melanoma tumor induction, C57BL/6NCrl mice were subcutaneously (s.c.) inoculated in the right flank with 1 × 10^5^ viable B16-F10 cells. For the breast cancer murine model, 1 × 10^4^ viable 4T1 cells were s.c. injected into the right mammary fat pad of BALB/cAnNCrl mice. To evaluate the effect of treatments on tumor growth, 5 days after tumor cells inoculation, 8 mice per group were treated with 75 mg/Kg (B16-F10 model) or 18.7 mg/Kg (4T1 model) body weight of P2Et extract, 142.5 mg/Kg body weight of *Tillandsia usneoides* extract (4T1 and B16-F10 models), or PBS (negative control) two times per week. To ensure low toxicity, P2Et and *T. usneoides* therapeutic dose was determined as fourfold lower than the LD_50_ estimation [21,22]. In all experimental settings, the size of the tumors was assessed three times per week with Vernier calipers, and the volume was calculated according to the formula V (mm^3^) = L (major axis) × W^2^ (minor axis)/2 [23]. Mice were euthanized by CO_2_ inhalation, and then spleen, tumor-draining lymph nodes (TDLN), and tumor were removed and processed. In addition, in the 4T1 breast cancer model, where metastases are clearly visible, the appearance of these in different organs was evaluated. The number of organs with macrometastasis was reported when small growth masses were observed at necropsy.

### 2.13. Evaluation of Immune Populations by Flow Cytometry

Briefly, 1 × 10^6^ cells were stained with LIVE/DEAD Fixable Aqua for 20 min in dark conditions at room temperature. After washing with PBS 2% FBS, the cells were stained for 30 min at 4 °C in dark conditions with the surface antibodies at a final concentration of 1 µg/mL according to the designed multicolor panels. To identify regulatory T cells, cells previously stained with LIVE/DEAD Fixable Aqua, anti-CD45, anti-CD3, anti-CD4, and anti-CD25 were fixed and permeabilized using the True Nuclear Transcription Factor Buffer Set (Biolegend) according to the manufacturer’s instructions. Then, cells were stained with anti-FoxP3 and anti-CTLA-4 antibodies for 30 min at room temperature in the dark, washed, and resuspended. The cells were acquired by flow cytometry using the Cytek Aurora Cytometer (Cytek Biosciences, Fremont, CA, USA), and the results were subsequently analyzed using FlowJo v10.8.1 software (BD Life Sciences). For analysis of dimensionality reduction, the OMIQ platform (Accessed on 25 March 2022, https://www.omiq.ai/) was used. Single live CD45 cells for each file were concatenated for analysis by opt-SNE dimensionality reduction followed by a comparison of each group in the concatenated file to identify the proportions of each population.

### 2.14. Evaluation of the Immune Response by Flow Cytometry

Splenocytes were stimulated with phorbol 12-myristate 13-acetate (PMA) and ionomycin for 6 h and the last 5 h of culture were performed with 1 μg/mL brefeldin A (BD Pharmingen). Cells of 1 × 10^6^ were incubated with LIVE/DEAD Fixable Aqua. Then, the cells were stained with anti-CD45, anti-CD3, anti-CD4, and anti-CD8 antibodies for 30 min at 4 °C in the dark. Later, the cells were washed, fixed, and permeabilized for final staining with anti-IFNγ, anti-TNFα, anti-IL-2, anti-perforin, and anti-granzyme B. Finally, the cells were washed and resuspended in PBS. Cells were acquired through flow cytometry using the Cytek Aurora Cytometer (Cytek Biosciences) and the results were subsequently analyzed using FlowJo v10.8.1 software (BD Life Sciences). Multifunctional analyses were performed using a Boolean gating strategy. The data are presented using Pestle v2.0 and SPICE v6.1 software (the National Institutes of Health, Bethesda, MD, USA) [24].

### 2.15. Statistical Analysis

Comparison between two groups was calculated using the Mann–Whitney U test. The Kruskal–Wallis test with Dunn’s post-test for multiple comparisons was used to evaluate differences between more than two groups. Differences were considered statistically significant when *p* < 0.05. Statistical analyses were performed by the GraphPad Prism version 8.1.1 for Mac OS X statistics software (GraphPad Software).

## 3. Results

### 3.1. Chromatographic Analysis

The chemical analysis by UPLC–PDA of the crude extract of *T. usneoides* at different wavelengths showed the presence of several peaks, with the major ones observed in Rt = 6.0 and 9.0 min (Figure 1). According to the maxima absorption in the UV spectra, the main peaks of the chromatogram are related to phenolic compounds, especially phenolic acids and flavonoids. The identification of the metabolites present in the extract is currently being carried out.

### 3.2. T. usneoides Extract Has Cytotoxic Activity, Induces Apoptosis, and Decreases the Proliferation of 4T1 and B16-F10 Cells

The cytotoxicity of *T. usneoides* extract was evaluated by MTT assay. Tumor cells were treated with different concentrations of the extract for 48 h. The extract reduced 4T1 and B16-F10 cell viability in a dose-dependent manner with an IC_50_ of 48.95 ± 3.74 µg/mL and 48.35 ± 5.53 µg/mL, respectively (Appendix A). To determine cell death mechanisms, cells were treated with the *T. usneoides* extract during 24 h, labeled with Annexin V and PI, and then, analyzed by flow cytometry. It was noted that 4T1 cells treated with both concentrations of the *T. usneoides* extract presented a significant increase in the frequency of apoptotic cells, as observed for the positive control doxorubicin, in a concentration-dependent manner (Figure 2A,B). The same was observed for the B16-F10 cells treated with the *T. usneoides* extract, being more sensitive to the extract and doxorubicin compared with 4T1 cells (Figure 2C,D). These results were correlated with a delay in the proliferative capacity, still observed at 1/5 of the IC_50_ and mainly at 12 h both in 4T1 and B16-F10 cells after treatment with the extract, although more significantly in 4T1 cells (Figure 2E,F).

### 3.3. T. usneoides Increases ROS in Both Cell Lines but Only Modifies Energetic Metabolisms in B16-F10 Cells

The modulation of intracellular ROS levels may be related to the cytotoxic activity of chemotherapeutic agents. We investigated if *T. usneoides* extract induces changes in ROS production using the H2DCFDA probe by flow cytometry. In fact, the *T. usneoides* extract induces an increase in ROS in 4T1 cells after 6 h of treatment that tends to decrease over time, while in B16-F10 it induces a significant time-dependent increase in ROS (Figure 2G). In both cases, the pro-oxidant effect was higher with the IC50 of the extract compared with the IC50/2 (Figure 2G). Despite the antagonistic behavior regarding the control of ROS in both cell lines, we evidenced a reduction in the proliferative capacity of the cells, which suggests that the mechanisms involved in the control of proliferation may be different, and may be conditioned to the intracellular microenvironment (Figure 2E,F) of the pro-oxidant (doxorubicin) and antioxidant (P2Et) (Figure 2G) previously studied in our group [25]. The increase in intracellular ROS is frequently accompanied by mitochondrial membrane depolarization and induction of caspase 3-dependent apoptosis. Interestingly, *T. usneoides* extract does not induce mitochondrial membrane depolarization (Appendix A), so its activity may be due to other mechanisms related to the control of energy metabolism. Given that glucose consumption by tumor cells constitutes an important source of energy, we evaluated the effect of the extract on glucose uptake using the 2-NBDG probe in both cell lines. Surprisingly, *T. usneoides* did not induce significant changes in glucose consumption in 4T1 cells, however, in B16-F10 it was observed that the extract induces an increase mainly at 12 h (Figure 2H). This finding is interesting since a significant hypoglycemic effect has been reported [9], but not an increase in the glucose uptake, in this case on tumor cells. Treatment with rotenone made it possible to demonstrate the difference in the intracellular microenvironment of each tumor population. In fact, rotenone increases glucose consumption, since it alters mitochondrial respiration by inhibiting complex I of the respiratory chain, decreasing the production of ATP [26]. While in 4T1 an increase in glucose consumption was observed after 12 h of rotenone treatment, in B16-F10 a decrease was observed (Figure 2H), suggesting differences in mitochondrial sensitivity to this disruptor and possibly a different metabolic plasticity of each tumor model.

### 3.4. The T. usneoides Extract Delays 4T1 Breast Cancer Tumor Growth

In order to evaluate whether the differences observed at the cellular level were manifested in a different sensitivity of each tumor to in vivo treatment, we evaluated the effect of the *T. usneoides* extract on the control of tumor growth (Figure 3A). Cells of 4T1 or B16-F10 were transplanted to BALB/c and C57BL/6 mice respectively and when tumors were established after 5 days, mice were treated with 142.5 mg/Kg body weight of *T. usneoides* extract previously calculated in an acute toxicity test (described in Section 2), P2Et extract (positive control) [27,28], or PBS (negative control). In the 4T1 breast cancer model, animals treated with *T. usneoides* displayed a significant delay in tumor progression, even greater than our positive control P2Et (Figure 3B; Appendix A). In addition, *T. usneoides* treatment reduced the number of mice with macrometastasis by 56% compared with the PBS group (Figure 3C,D; Appendix A). However, in the melanoma B16-F10 model, *T. usneoides* extract did not show an effect on tumor growth (Figure 3E; (Appendix A)) despite the greater sensitivity of these cells to in vitro treatment with the extract. In both models, we confirmed that P2Et extract significantly delayed tumor growth compared with the PBS group as previously shown (Figure 3B,E).

### 3.5. T. usneoides Extract Modulates the Tumor Microenvironment in 4T1 Breast Cancer Tumor

To evaluate if the tumor microenvironment is modulated by the treatment, we compared the frequency of some intratumoral immune cells present in 4T1 and B16-F10 models. In the 4T1 model, animals treated with P2Et and *T. usneoides* extracts had a significantly higher proportion of CD45^+^ cells compared with the control group (Figure 4A). An overview of the main tumor-infiltrating immune cells identified by flow cytometry showed a higher proportion of conventional dendritic cells (cDCs) and CD8*α* DCs in the *T. usneoides* group compared with the control group (Figure 4B). Simultaneously, the frequency of PMN-MDSC-like cells (LC) decreased after *T. usneoides* treatment (Figure 4B). The t-SNE analysis of tumor tissues from mice after different treatments indicated a decrease in MDSC-LC and an increase in DCs in animals treated with both extracts, compared with the PBS group (Figure 4C). Add to this, mice treated with P2Et and *T. usneoides* had an increased frequency of activated CD8^+^ CD44^+^ T cells (Figure 4D), suggesting that *T. usneoides* extract, as well as P2Et, as previously shown [27,28], may enhance the antigenic presentation and limit the infiltration of MDSC-LC in the TME. In the B16-F10 model, we only observed the increase in tumor-infiltrating immune cells in mice treated with P2Et, but not with *T. usneoides* (Figure 4E). However, the fine analysis of tumor-infiltrating immune populations let us demonstrate a decrease in the frequency of Treg and M-MDSC-LC in mice treated with *T. usneoides* compared with the PBS group (Figure 4F), but no differences in the frequency of DCs (Figure 4F,G) and the activated CD8^+^ T cells (Figure 4H). The t-SNE analysis of tumor tissues from mice after different treatments did not indicate significant changes between the *T. usneoides* and PBS group (Figure 4G). As previously reported, P2Et induced an increase in CD8^+^ T cells, cDCs, and CD8*α*^+^ DCs, and a decrease in Treg and PMN-MDSC-LC (Figure 4F) [22,27,28].

### 3.6. T. usneoides Treatment Modulates the Immune Response in 4T1 Tumor-Draining Lymph Nodes

The evaluation of immune cell populations in 4T1 tumor-draining lymph nodes (TDLNs) showed a higher frequency of CD3^+^ and CD4^+^ cells while a lower frequency of suppressor cells such as Tregs and M-MDSC-LC (Figure 5A). Likewise, these results are related to an increase in the frequency of activated CD8^+^ T cells in comparison with the PBS group (Figure 5B). These results suggest that the treatment with *T. usneoides* extract could be favoring the activation of T cells that subsequently migrate to the tumor. Conversely, in the melanoma model, only a decrease in PMN-MDSC-LC was found (Figure 5C) and no differences were found in the percentage of activated CD8^+^ T cells (Figure 5D). As expected, in both models the P2Et extract increased the frequency of T cells and decreased the suppressor cells (Figure 5A,C).

### 3.7. T. usneoides Treatment Enhances Functional Activity of T Cells in 4T1 Breast Cancer but Not in Melanoma

Based on the previous results, we evaluated if the modulation of immune cell populations in tumor and tumor-draining lymph nodes correlates with a better quality of the response of T cells. For this, the production of cytokines was evaluated individually and simultaneously in T cells from spleen of both murine models. In the 4T1 breast cancer model, a higher frequency of CD4^+^ (Figure 6A) and CD8^+^ T cells (Figure 6B) producing IFNγ, TNFα, or IL-2 was found in mice treated with *T. usneoides*. When cytokines production was simultaneously evaluated, mice treated with *T. usneoides* showed a higher frequency of polyfunctional CD4^+^ (Figure 6C) and CD8^+^ (Figure 6B) T cells compared with the control group. As expected, these results were also found in animals treated with P2Et extract (Figure 6). In the B16-F10 murine model, an increase in the frequency of CD4^+^ T cells producing IFNγ, TNFα, IL-2, granzyme B, or perforin (Figure 7A), and CD8^+^ T cells producing IFNγ, TNFα, IL-2, or perforin (Figure 7B) was observed only with the P2Et treatment but not with the *T. usneoides* extract. In the same way, polyfunctionality was found in T cells from animals treated with P2Et but not with *T. usneoides* extract (Figure 7C,D). Taken together, these results suggest that the success of the *T. usneoides* extract in controlling 4T1 tumor growth may be due, in part, to the intrinsic sensitivity of 4T1 acting on the tumor microenvironment, favoring the activation and generation of an effective adaptive antitumor immune response.

## 4. Discussion

Many plant-derived natural products exhibit activity in murine tumor models [29,30]. In fact, many antitumor agents have been isolated from plants, but to date the concept that natural products can function in isolation does not consider tumor complexity. The tumor is a sum of genetic and metabolic alterations that are intrinsic to the tumor cell, and in fact, the relationship between the tumor and the microenvironment can play for or against tumor survival [25,26].

It is not surprising that in this work we observed that a complex extract from a plant had differential effects in two murine tumor models. The extract of *T. usneoides*, despite presenting in vitro cytotoxic activity on both 4T1 breast cancer and B16 melanoma tumor cells, only shows activity in vivo on the 4T1 model. The induction of cell death in both models seems to occur by different mechanisms. While in 4T1, the *T. usneoides* extract presents a clear antioxidant activity and a reduction in glucose uptake; in B16-F10 the effect over time is inverse for both measurements, even though B16-F10 cells are more sensitive to the induction of cytotoxicity by the extract of *T. usneoides*.

For several decades, it has been shown that the type of death in tumor cells determines what happens around them. It has been postulated that death by apoptosis is desired in antitumor drug discovery because of the relation with the immunogenic cell death, since death by necrosis can increase a local proinflammatory environment with deleterious effects on the tumor microenvironment [31]. In our case, in both cells, although the *T. usneoides* extract induces death by apoptosis, tumor control is different and the factors involved seem to be related to a better induction of the immune response, when in this case, the extract is capable of reducing glucose consumption in conjunction with intracellular oxidation.

In 4T1 tumor-bearing mice, a clear decrease in tumor and metastasis was observed, accompanied by a decrease in intratumoral PMN-MDSC-LC, an increase in antigen-presenting DCs, and a significant decrease in regulatory T cells in TDLNs. In contrast, in B16-F10, although changes were observed in the intratumoral distribution of some cells of the immune response, these did not have an effect on tumor control.

MDSC are immature myeloid cells that are increased in patients with cancer, trauma, and chronic inflammation, possibly due to the production of GM-CSF and other cytokines that promote their expansion. The production of anti-inflammatory cytokines by MDSCs, such as IL-10 or TGF-β, inhibits the response of M1 macrophages, increasing M2, which promote angiogenesis and metastasis. Several flavonoids, triterpenoids, retinoids, curcuminoids, onionins, and withanolides, among others, have shown activity on MDSC, inhibiting their intratumoral accumulation and favoring their differentiation, which decreases their immunosuppressive capacity. Although the mechanisms by which this regulation occurs have not been clearly established to date, the ability of many of these compounds to inhibit NFkB translocation and to activate AMP-activated protein kinase (AMPK) could partly explain their function [32,33]. Receptors such as retinoic acid receptor-related orphan receptors (RORs) have been implicated in the function of MDSC and may also be the target of some natural products [34], as well as the non-receptor tyrosine kinases (NRTK) expressed in hematopoietic cells, opening an interesting field in the search for natural modulators of suppression in cancer [35].

The antitumor activity of P2Et has already been previously published in both models [20,21,22,27,28]; however, in this work clear differences in the tumor microenvironment of B16-F10 vs. 4T1 without the influence of treatment are evidenced, such as the large amount of PMN-MDSC-LC in the 4T1 murine model that contrasts with the highest frequency of M-MDSC-LC in the B16-F10 murine model. Likewise, the basal frequency and the response to therapy of CD4^+^ and CD8^+^ T cells was greater in 4T1 than in the B16-F10 murine model, suggesting that 4T1 is a hotter tumor than B16-F10, which may be conditioned by the intrinsic differences of each model and its relationship with the tumor microenvironment. A hot TME can be described as proinflammatory and antitumor, characterized by high TIL and dendritic cell counts, and low presence of immunosuppressive factors such as MDSC and Tregs [36], which correlate with our findings in 4T1 tumor-bearing mice treated with *T. usneoides* extract. In contrast, cold TME is generally immunosuppressive and promotes tumor growth.

Recently, Cohen et al. [37] showed molecular and phenotypic differences in response to treatment with non-ablative pulse-focused ultrasound (pFUS) in 4T1 vs. B16-F10 murine models and suggests that this discrepancy is due to the profound heterogeneity of the tumors. Our group has also previously shown that although P2Et has antitumor activity in 4T1 breast cancer and B16-F10 melanoma models, the mechanisms by which this control is exerted are different and, in addition, the proinflammatory state of the animals influences their response to treatment [27].

Several activities have been reported for *T. usneoides*, including microbicide due to the presence of a flavonol-type glycoside; hypoglycemic activity attributed to 3-hydroxy-3-methylglutaric acid (HMG), and antiviral activity due to the presence of polyphenols. In addition, other research has reported *T. usneoides* as anti-hypertensive and active in rheumatism, hemorrhoids, cholagogue, diuretic, renal and ophthalmic illnesses [38]. Other *Tillandsia* genera exhibit anti-inflammatory and cytotoxic activity in vitro against different tumor cells [39,40,41]. Crude ethanolic extract of *T. recurvata* inhibits the growth of Kaposi’s sarcoma in mice [41] and its activity has been attributed to the presence of 1,3-di-O-Cinnamoyl-glycerol and (E)-3-(cinnamoyloxy)-2-hydroxypropyl 3-(3,4-dimethoxyphenyl)acrylate [40]. A flavone (5,3′-dihydroxy-6,7,8,4′-tetramethoxyflavanone) with broad in vitro activity against tumor cells has been isolated and could be useful in glioblastoma multiforme and acute myeloid leukemia in addition to other cancers [39].

Pharmacological inhibition of glucose uptake has been described as a therapeutic target for cancer. To date, natural products have become more valuable in the discovery of drugs. Interestingly, with the *T. usneoides* extract, we only found a modulation with a tendency to increase the glucose uptake in the B16-F10 model. Previously, it has been described that *T. usneoides* stimulates insulin secretion in the absence of changes in the intracellular concentration of Ca^2+^ in RINm5F cells, generating a hypoglycemic effect [42]. The increase in the glucose uptake must occur through one of the following: an increase in the total expression of the transporter, recruitment of new transporters into the plasma membrane, or activation of the transporters already residing in the plasma membrane for the production of biosynthetic precursors needed to support rapid proliferation. This metabolic change provides the necessary substrates for tumor cell proliferation, which could explain the in vivo result in the melanoma model.

The investigation of new antitumor therapies has focused on the isolation of compounds with direct antitumor activity, ignoring the fact that the tumor is a complex system that interacts with its host to survive [43]. Although tumor elimination is the basis of antitumor therapy, the high toxicity due to the low specificity of antitumor agents means that many of them cannot be used in clinical trials. In this study, we show that the complex mixtures obtained from *T. usneoides* can present a selective antitumor activity that not only reduces tumor size, but also favors the activation of the immune response. The use of omics tools through network pharmacology will make it possible to understand these differences, allowing a rational development of polymolecular medicines, such as standardized plant extracts.

## 5. Conclusions

*T. usneoides* extract antagonistically regulates tumor metabolism of 4T1 vs. B16-F10. While in the first, glucose consumption decreases as well as intracellular ROS, in the second, glucose consumption increases as well as ROS. These differences seem to have a significant impact on the tumor microenvironment, allowing the tumor to shrink in 4T1, while not in B16-F10. This result is apparently related to the potentiation of an effective antitumor immune response. The mechanisms related to the intrinsic sensitivity of the different tumor cells to each of these compounds or extracts could be related to differences in their metabolic plasticity. It is also surprising that these differences are observed in response to treatment with a complex extract of *T. usneoides*, suggesting that the development of polymolecular medicines from plants may be possible. These data, analyzed by network pharmacology, would make it possible to understand the mechanisms involved in differential sensitivity in vivo, and to scientifically validate traditional knowledge, taking advantage of it for the rational discovery of mixtures of metabolites with antitumor activity.

## Figures and Tables

**Figure 1 cancers-14-05383-f001:**
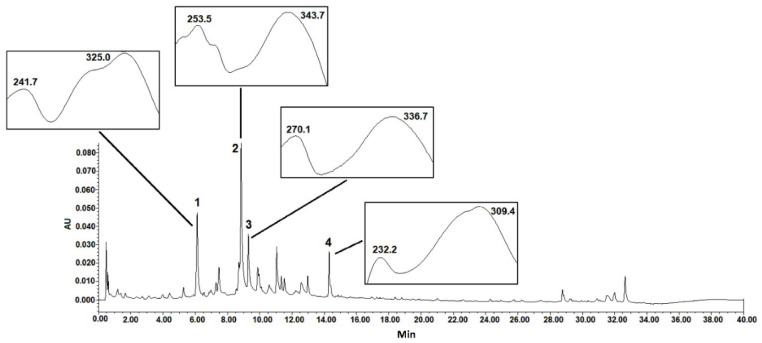
Chromatographic analysis of *T. usneoides* ethanolic extract at 254 nm and UV spectra. UPLC–PDA analysis was performed with an Acquity UPLC H-class (Waters, Milford, MA, USA) and a Phenomenex Kinetex C 18 column (100 × 2.1 mm, 1.7 µm).

**Figure 2 cancers-14-05383-f002:**
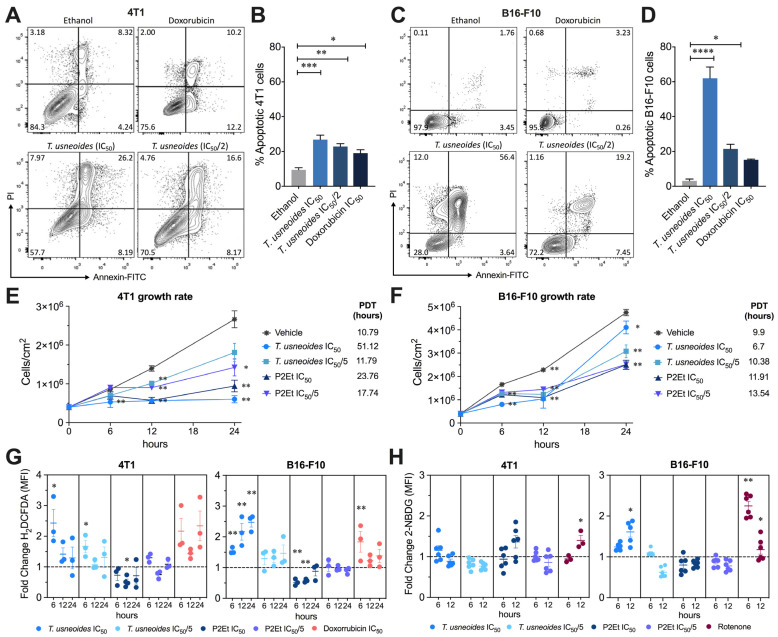
*T. usneoides* extract-induced apoptosis on 4T1 cells. (**A**). Representative contour plots of 4T1 cells incubated with IC_50_ (48.95 µg/mL) and IC_50_/2 (24.47 µg/mL) of *T. usneoides* extract, negative control (ethanol), or control positive (doxorubicin) for 24h. In representative flow cytometry analysis, necrotic (Annexin V^−^, PI^+^), late apoptotic (Annexin V^+^, PI^+^), early apoptotic (Annexin V^+^, PI^−^), and viable (Annexin V^−^, PI^−^) cells were indicated. (**B**). Frequency of apoptotic 4T1 cells (sum of early and late apoptosis) expressed as mean ± SEM for three independent experiments. (**C**). Representative contour plots of B16-F10 cells incubated with IC_50_ (48.35 µg/mL) and IC_50_/2 (24.18 µg/mL) of *T. usneoides* extract, negative control (ethanol), or control positive (doxorubicin) for 24 h. (**D**). Frequency of apoptotic B16-F10 cells (sum of early and late apoptosis) expressed as mean ± SEM for three independent experiments. (**E**). 4T1 and (**F**) B16-F10 cell count per cm^2^ after treatment with IC_50_ and IC_50_/5 of *T. usneoides* extract, IC_50_ and IC_50_/5 of P2Et extract, or vehicle (ethanol) for 0 h, 6 h, 12 h, and 24 h. Population doubling times (PDT) are shown for each treatment. (**G**). Fold change of H_2_DCFDA MFI after the treatments with IC_50_ and IC_50_/5 of *T. usneoides* extract, IC_50_ and IC_50_/5 of P2Et extract, or doxorubicin IC_50_ (positive control) for 6 h, 12 h, and 24 h in both cell lines. (**H**). Fold change of 2-NBDG MFI after treatments with IC_50_ and IC_50_/5 of *T. usneoides* extract, IC_50_ and IC_50_/5 of P2Et extract, or rotenone (positive control) for 6 h and 12 h. In all cases, fold change was determined using the MFI of each treatment relative to the vehicle (ethanol or DMSO). Data from three independent experiments are shown. * *p* < 0.05; ** *p* < 0.01; *** *p* < 0.001; **** *p* < 0.0001.

**Figure 3 cancers-14-05383-f003:**
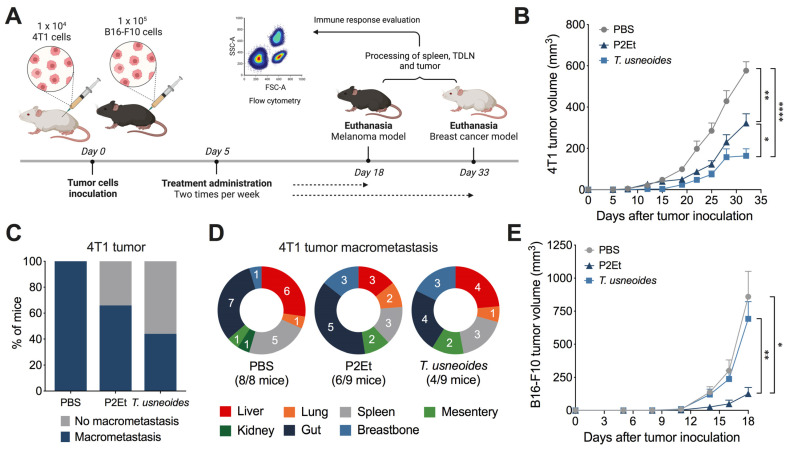
In vivo *T. usneoides* treatment delays breast tumor growth. (**A**). Experimental design to evaluate the antitumor effect of *T. usneoides* extract in BALB/c or C57BL/6 mice bearing 4T1 or B16-F10 tumors, respectively. Tumor was established by injection of 4T1 or B16-F10 cells; then, 5 days after tumor cell injection, treatments were administrated two times per week until the end of the experiment. (**B**). Tumor volume in 4T1 tumor-bearing mice treated with each treatment. (**C**). Bars showing the percentage of mice that developed macrometastasis in 4T1 breast cancer model. (**D**). Distribution of multi-organ metastasis of 4T1 tumors for all groups. (**E**). Tumor volume in B16-F10 tumor-bearing mice treated with each treatment. The numbers on the pies show the mice with macrometastasis and numbers in parenthesis correspond to the total number of mice with metastases. The *p* values were calculated using Kruskal–Wallis and Dunn’s post test for multiple comparisons. * *p* < 0.05, ** *p* < 0.01, **** *p* < 0.0001.

**Figure 4 cancers-14-05383-f004:**
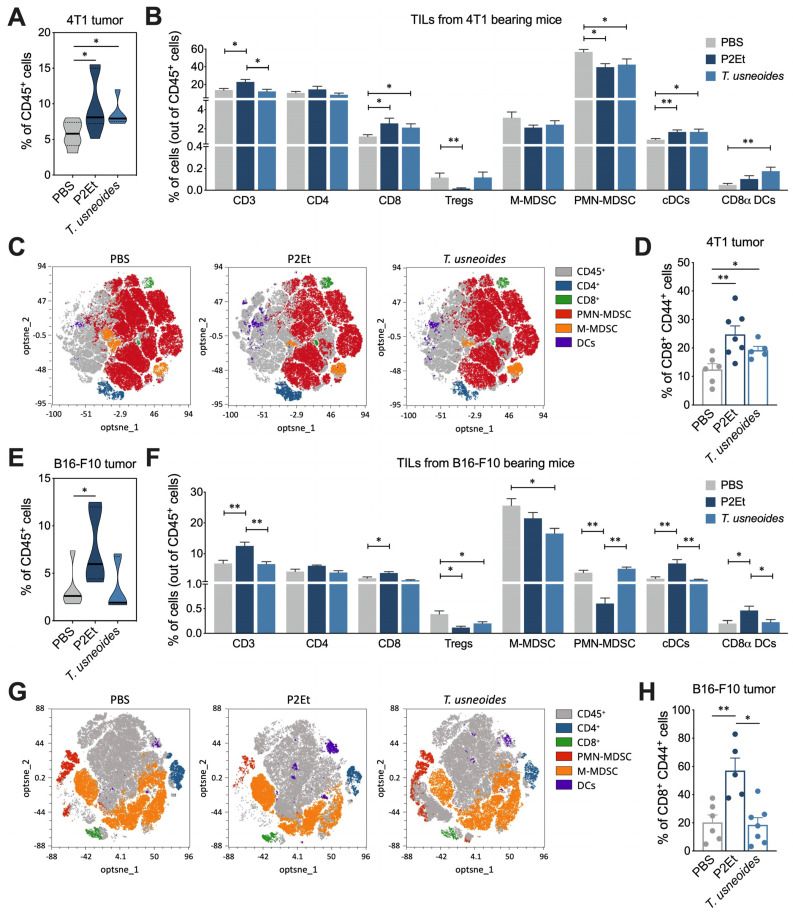
*T. usneoides* extract modulates the tumor microenvironment in 4T1 breast cancer tumor. (**A**). Frequency of 4T1 intratumor CD45^+^ cells in mice treated with P2Et, *T. usneoides*, or PBS (control). (**B**). Overview of the immune cell composition in the 4T1 TME shown in the percentage of cells (out of CD45^+^ cells) on a per-mouse basis. (**C**). T-distributed stochastic neighbor embedding (t-SNE) visualization of clustering of some immune subpopulations from the 4T1 tumor detected by flow cytometry; each dot corresponds to one single cell. (**D**). Frequency of activated CD8^+^ T cells. (**E**). Frequency of B16-F10 intratumor CD45^+^ cells in mice groups. (**F**). Overview of the immune cell composition in the B16-F10 TME shown in the percentage of cells (out of CD45^+^ cells). (**G**). T-distributed stochastic neighbor embedding (t-SNE) visualization of clustering of some immune subpopulations from the B16-F10 tumor detected by flow cytometry; each dot corresponds to one single cell. (**H**). Frequency of activated CD8^+^ T cells. In all cases, data are represented as the mean ± SEM. The *p* values were calculated using the Mann–Whitney U test. * *p* < 0.05, ** *p* < 0.01.

**Figure 5 cancers-14-05383-f005:**
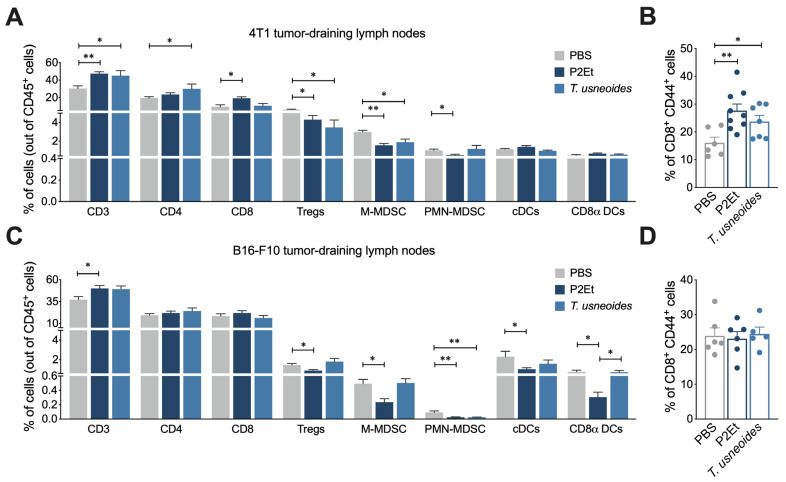
Treatment of *T. usneoides* in vivo modulates the immune response in lymph nodes. (**A**). Overview of the immune cell composition in lymph nodes from 4T1 tumor bearing mice shown in percentage of cells on a per-mouse basis. (**B**). Frequency of activated CD8^+^ T cells. (**C**). Overview of the immune cell composition in lymph nodes from B16-F10 tumor bearing mice shown in percentage of cells on a per-mouse basis. (**D**). Frequency of activated CD8^+^ T cells. In all cases, data are represented as the mean ± SEM. The p values were calculated using the Mann–Whitney U test. * *p* < 0.05, ** *p* < 0.01.

**Figure 6 cancers-14-05383-f006:**
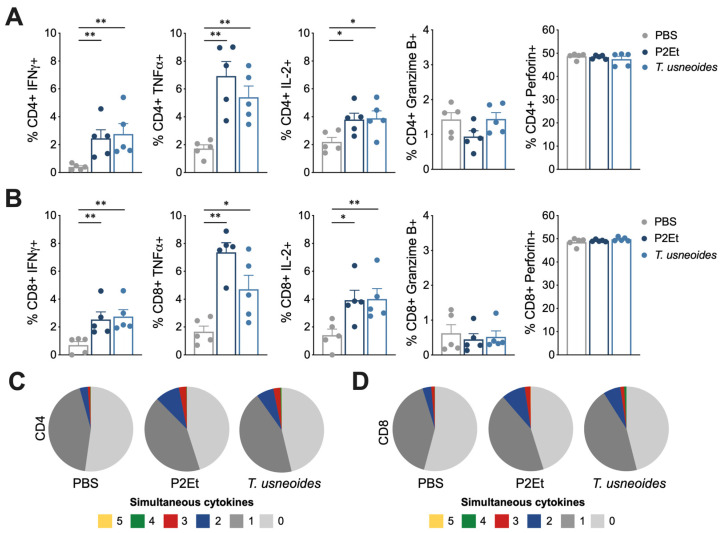
*T. usneoides* extract modulate functional activity of T cells in the breast cancer model. (**A**). Frequency of CD4^+^ T cells from spleen producing IFNγ, TNFα, IL-2, Granzyme B, and Perforin following stimulation with PMA/ionomycin (P/I). (**B**). Frequency of CD8^+^ T cells from spleen producing IFNγ, TNFα, IL-2, Granzyme B, and Perforin following stimulation with P/I. Polyfunctional activity of CD4^+^ (**C**) or CD8^+^ (**D**) T cells from spleen, following stimulation with P/I, from each group of treatment. The functional profiles are grouped and color-coded according to the number of functions, as shown in the pie charts. * *p* < 0.05, ** *p* < 0.01.

**Figure 7 cancers-14-05383-f007:**
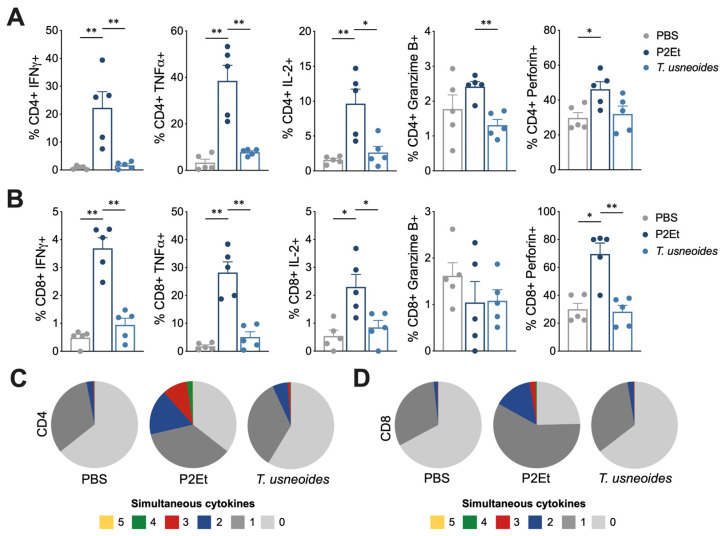
*T. usneoides* extract does not modulate functional activity of T cells in the melanoma model. (**A**). Frequency of CD4^+^ T cells from spleen producing IFNγ, TNFα, IL-2, Granzyme B, and Perforin following stimulation with PMA/ionomycin (P/I). (**B**). Frequency of CD8^+^ T cells from spleen producing IFNγ, TNFα, IL-2, Granzyme B, and Perforin following stimulation with P/I. Polyfunctional activity of CD4^+^ (**C**) or CD8^+^ (**D**) T cells from spleen, following stimulation with P/I, from each group of treatment. The functional profiles are grouped and color-coded according to the number of functions, as shown in the pie charts. * *p* < 0.05, ** *p* < 0.01.

## Data Availability

The data presented in this study are available in this article (and Appendix A).

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
