# Peer review of "Tillandsia usneoides Extract Decreases the Primary Tumor in a Murine Breast Cancer Model but Not in Melanoma"

_cancers, 2022, doi:10.3390/cancers14215383_

Round 1

Reviewer 1 Report

The authors treat in this work a topic of great actuality and relevance as it is the Discovery of natural products as potential leads to antitumor drugs in the study of medicinal plants. They propose as a starting point a solid hypothesis on the use of Tillandsia usneoides as an antitumor drug. The methodological proposal is adequate and robust, covering different methodologies to be able to respond to the proposed hypothesis. The results obtained are robust and support the conclusions reached.

A very positive point of the work is that it takes into account that the antitumor activity of a compound may not affect the tumor cells per se, but rather the tumor microenvironment, for which they analyze the variations in the immune system.

Some small changes that help to better follow the work done:

(1) To facilitate the understanding of the bar graphs, and considering that the work may be printed in B/W, if it would be possible to change the dark blue and light blue colors for two that are better differentiated, it could be fine. 

(2) Figure 2 could be better located after result point 3.2 

(3) Same as for bar chart, the line chart (for ex. Fig 3B and similar to it)  would be better understood if the points for each condition have a different shape (circle, square, cross,... for example)

(4) To my pleasant surprise, the authors have used female specimens for their animal models. As the authors will know, there are few works in which female models are used due to a gender bias in the research.

I understand that the choice of the sex of the mice has been due to the use of the breast cancer cell line. However, I am not sure that it has been a good choice in the case of melanoma, since the frequency is higher in men and also the cell line used derives from a male mouse (that is, they are XY cells), could this somehow influence the results obtained? It would be good to repeat some test with male mice to see if the results are replicated or differ. It would be very useful to have this information since we could be facing a case of treatment that works better for one sex than for another. 

In any case, it is convenient to specify in the results and in the discussion that the results obtained apply to a female model and not extrapolate it to the general population, since it could be that the results are not the same for a male model.

Author Response

We appreciate and welcome your review, as well as all of your comments and suggestions. Here is an answer to each of them:

Point 1: To facilitate the understanding of the bar graphs, and considering that the work may be printed in B/W, if it would be possible to change the dark blue and light blue colors for two that are better differentiated, it could be fine. 

Response 1: We appreciate the suggestion, however, given that in some figures many variables are presented that would not be easily differentiated in black and white, it is contemplated to pay for the publication in color if it is accepted.

Point 2: Figure 2 could be better located after result point 3.2

Response 2: According to the recommendation the figure was moved after result point 3.2

Point 3: Same as for bar chart, the line chart (for ex. Fig 3B and similar to it) would be better understood if the points for each condition have a different shape (circle, square, cross,... for example)

Response 3: Based on the reviewer's recommendation, Figures 3B and 3E were modified.

Point 4: To my pleasant surprise, the authors have used female specimens for their animal models. As the authors will know, there are few works in which female models are used due to a gender bias in the research.

I understand that the choice of the sex of the mice has been due to the use of the breast cancer cell line. However, I am not sure that it has been a good choice in the case of melanoma, since the frequency is higher in men and also the cell line used derives from a male mouse (that is, they are XY cells), could this somehow influence the results obtained? It would be good to repeat some test with male mice to see if the results are replicated or differ. It would be very useful to have this information since we could be facing a case of treatment that works better for one sex than for another.

In any case, it is convenient to specify in the results and in the discussion that the results obtained apply to a female model and not extrapolate it to the general population, since it could be that the results are not the same for a male model.

Response 4: We understand your concern with the melanoma model. However, in the laboratory we have made the model with males and we have evaluated the effect of the P2Et extract, in this case our positive control, finding similar results to those found when the model is made with females. We believe that this is an important point to evaluate in the future and, thus, confirm that the effect of the Tillandsia usneoides extract is the same in males and females.

Reviewer 2 Report

Reviewer Comments

Journal:  Cancers

Manuscript ID – Cancers-1958615

Title: Tillandsia usneoides extract decreases the primary tumor in a 2 murine breast cancer model but not in melanoma

Author: Paola Lasso, Laura Rojas, Cindy Arévalo, Claudia Urueña, Natalia Murillo, Alfonso Barreto, Geison M Costa and Susana Fiorentino.

 Review:

 Manuscript was well written; background was clearly explained the importance of this study. Material and Methods are clearly explained, and result are well written. In future, if author focused on main ingredient present in the T. usneoides will lead to the future direction. Other important aspect is, there is no mechanistic study has performed in this manuscript, in future author focused on mechanistic study how extract is inhibiting the tumor growth, that would be very helpful for the treatment. Overall, this paper is very interesting paper focusing on natural product for cancer treatment.

 Major Comments:

1)                  In paragaraph, 3.2-line 266-268, Author mentioned that T. usneoides extract delayed the proliferation, but in Figure 2E and F shows increase in cell number after T. usneoides treatment, and P2Et, data is little confusing. Author needs to represent the data in line profile or different method to make his point clear.

2)                  In Figure-2, Data was clearly shown that T. usneoides induces apoptosis, delay in proliferation proliferation, and increase in ROS based on IC50 concentration of T. usneoides. But there is no data showing the time and dose dependent curve to decide the IC-50 concentrations.  I would recommend, author to show the time and dose dependent curve for both T. usneoides and P2Et.

3)                  i) In Figure-3, Author mentioned that T. usneoides extract significantly reduces the tumor volume and macrometastasis in 4T1 injected mice when compared to control. If author showed the image of the tumor size with and without extract will give more support to the data (Figure 3B&E).

ii) Author did not explain how he detected macrometastasis or multiorgan metastasis and no data like tissue staining, or tumor images were shown in the manuscript. Author needs to show metastatic site images to convince the Figure-3D data.

Minor Comments:

1)              In the Introduction, Author has mentioned in line 72, “Given the little evidence found in the literature, related to the antitumor activity of the T. usneoides species but based on its biological closeness to T. recurvata, widely studied [18-20]“ .  Author needs to explain the rationale of using T. usneoides when compared to T. recurvata. Even, T. recurvata was not well studied as anti-cancer property.

2)                  In this manuscript, author used P2Et as a positive control for T. usneoides. I would recommend author to perform with T. recurvata extract to show T. usneoides has a better anti-tumor activity.

3)                  In Figure 1, author have shown the chemical analysis of T. usneoides, if author compared this data with T. recurvata extract would be more strengthen the paper.  Previous literature showed T. recurvata extract was performed with methanol or ethanol or chloroform to test in various cancer cell line (1-6). Author needs to explain why they selected the methanol extraction and discuss more about the ingredient present in the T. usneoides.

4)                  Figure 4-7, Author wonderfully performed the data analyzation. If author would have discussed more about, How T. usneoides regulate or modulate the immune response in the discussion section will give more impact to the paper.

 References:

1)      HLBT-100: a highly potent anti-cancer flavanone from Tillandsia recurvata (L.) L.Lowe HIC, Toyang NJ, Watson CT, Ayeah KN, Bryant J. Cancer Cell Int. 2017 Mar 7;17:38.

2) Antileukemic activity of Tillandsia recurvata and some of its cycloartanes.Lowe HI, Toyang NJ, Watson CT, Ayeah KN, Bryant J. Anticancer Res. 2014 Jul;34(7):3505-9.

3) In vitro and in vivo anti-cancer effects of tillandsia recurvata (ball moss) from Jamaica.Lowe HI, Toyang NJ, Bryant J. West Indian Med J. 2013 Mar;62(3):177-80.

4)      In Vitro Anticancer Activity of the Crude Extract and two Dicinnamate Isolates from the Jamaican Ball Moss (Tillandsia Recurvata L.). Lowe HI, Toyang NJ, Watson C, Badal S, Bahado-Singh P, Bryant J. Am Int J Contemp Res. 2013 Jan;3(1):93-96.

5)      Cycloartane-3,24,25-triol inhibits MRCKα kinase and demonstrates promising anti prostate cancer activity in vitro. Lowe HI, Watson CT, Badal S, Toyang NJ, Bryant J.Cancer Cell Int. 2012 Nov 14;12(1):46.

6)      Kinase inhibition by the Jamaican ball moss, Tillandsia recurvata L.Lowe HI, Watson CT, Badal S, Toyang NJ, Bryant J. Anticancer Res. 2012 Oct;32(10):4419-22.

Author Response

We appreciate and welcome your review, as well as all of your comments and suggestions. Here is an answer to each of them:

Point 1:   In paragraph, 3.2-line 266-268, Author mentioned that T. usneoides extract delayed the proliferation, but in Figure 2E and F shows increase in cell number after T. usneoides treatment, and P2Et, data is little confusing. Author needs to represent the data in line profile or different method to make his point clear. 

Response 1: We agree with your suggestion; the data was analyzed in another way to better represent the result. The cell count per cm2 was included and, in addition, the doubling times of the cells treated with each treatment were included.

Point 2: In Figure-2, Data was clearly shown that T. usneoides induces apoptosis, delay in proliferation proliferation, and increase in ROS based on IC50 concentration of T. usneoides. But there is no data showing the time and dose dependent curve to decide the IC-50 concentrations. I would recommend, author to show the time and dose dependent curve for both T. usneoides and P2Et.

Response 2: We agree with the reviewer and added the results of MTT where we calculated the IC50 of Tillandsia usneoides at 48 hours. Considering these results, we choose the concentrations of each extract in order to test biological changes in a sublethal time and dose. Additionally, Figure 2 shows the data of each test with the IC50 and IC50/2 or IC50/5 of Tillandsia usneoides and P2Et, at the different times tested. For example, proliferation and ROS measurement were evaluated at 6, 12, and 24 hours, and glucose uptake at 6 and 12 hours. To complement the result, we include the IC50 determination curve in the supplementary results (Supplementary Figure 1).

Point 3:

i) In Figure-3, Author mentioned that T. usneoides extract significantly reduces the tumor volume and macrometastasis in 4T1 injected mice when compared to control. If author showed the image of the tumor size with and without extract will give more support to the data (Figure 3B&E).

ii) Author did not explain how he detected macrometastasis or multiorgan metastasis and no data like tissue staining, or tumor images were shown in the manuscript. Author needs to show metastatic site images to convince the Figure-3D data.

Response 3: Consistent with the suggestion, we include representative photos of tumor size and the presence of macrometastases in the supplementary results (Supplementary Figure 3). Additionally, we include a brief description in the paper of how macrometastasis was evaluated.

Point 4: In the Introduction, Author has mentioned in line 72, “Given the little evidence found in the literature, related to the antitumor activity of the T. usneoides species but based on its biological closeness to T. recurvata, widely studied [18-20]“. Author needs to explain the rationale of using T. usneoides when compared to T. recurvata. Even, T. recurvata was not well studied as anti-cancer property.

Response 4: We agree with the reviewer. For this reason, we have revised the wording of the sentence to make the specific interest in T. usneoides clearer.

Point 5: In this manuscript, author used P2Et as a positive control for T. usneoides. I would recommend author to perform with T. recurvata extract to show T. usneoides has a better anti-tumor activity.

Response 5: We thank the evaluator for the recommendation. We have been working on the study of Caesalpinia spinosa extract for 15 years and we are currently conducting a clinical study with the standardized extract of this plant. This is the reason why we use our P2Et extract as a positive control. Furthermore, we are currently working on 30 different species of our biodiversity, including T. recurvata. When we have the results of the chemical and biological characterization of the extract obtained from this plant, we will be able to share our results through a new publication, complementary to this one.

Point 6:   In Figure 1, author have shown the chemical analysis of T. usneoides, if author compared this data with T. recurvata extract would be more strengthen the paper. Previous literature showed T. recurvata extract was performed with methanol or ethanol or chloroform to test in various cancer cell line (1-6). Author needs to explain why they selected the methanol extraction and discuss more about the ingredient present in the T. usneoides.

Response 6: The extract of T. usneoides was obtained in ethanol, and this extraction method was chosen for its cytotoxic activity on the 4T1 and B16-F10 cell lines, and also for presenting the best selectivity index compared to other preparations evaluated. In addition, the ethanolic extract presented a variety of compounds with different polarities and, with respect to the other fractions, it is the one with the best extraction yield. In this paper we do not delve into the chemical part of the extract because these results are presented and discussed in a submitted article, entitled: “Chemical characterization and evaluation of biological activity on tumor cell lines of Tillandsia usneoides leaf extract”. 

We are actually tested the T. recurvata extract, and off course the results we get with this extract will be compared with T usneoides. It is not our intention to claim the role of T. usneoides on T. recurvata. In fact, our interest is to take advantage of this species that grows wild in one of the central regions of our country and that can be propagated in our hands more easily than Tillandsia recurvata.

Point 7:   Figure 4-7, Author wonderfully performed the data analyzation. If author would have discussed more about, How T. usneoides regulate or modulate the immune response in the discussion section will give more impact to the paper.

Response 7. We thank the reviewer for the comment. We added the next paragraph in the discussion:

MDSC are immature myeloid cells that are increased in patients with cancer, trauma and chronic inflammation, possibly due to the production of GM-CSF and other cytokines that promote their expansion. The production of anti-inflammatory cytokines by MDSCs, such as IL-10 or TGF-β, inhibits the response of M1 macrophages, increasing M2, which promote angiogenesis and metastasis. Several flavonoids, triterpenoids, retinoids, curcuminoids, onionins and withanolides, among others, have shown activity on MDSC, inhibiting their intratumoral accumulation and favoring their differentiation, which decreases their immunosuppressive capacity. Although the mechanisms by which this regulation occurs have not been clearly established to date, the ability of many of these compounds to inhibit NFkB translocation and to activate AMP-activated protein kinase (AMPK) could partly explain their function [1, 2]. Receptors such as Retinoic acid receptor-related orphan receptors (RORs), have been implicated in the function of MDSC and may also be the target of some natural products [3], as well as the Non-receptor tyrosine kinases (NRTK) expressed in hematopoietic cells, opening an interesting field in the search for natural modulators of suppression in cancer [4].

Reviewer 3 Report

The authors investigated the potential anti-tumour potential of a crude extract of Tillandsia usneoides, a plant commonly found in their country. I am happy with the approach they followed and the experimental designs they used for their studies. However, I have few questions to be answered, and put them as follows.

1. Is there use of the plant by traditional medicine practitioners or locals for treatment of cancer in your country?

2. How did you select 4T1 and B16-F10 cell lines for your study? did you test your extract against other cell lines? Given that your extract showed varying effects between the two cell lines, it would be logical to further investigate it against other cells, too.

3. I was confused and found it a bit annoying the way you reported IC50 values of your extract against 4T1 and B16-F10 cells, and it needs revision There is inconsistency, and hence, I would like to see the dose-response curve that you generated with Graphpad and the type of fit you used.

a) In section 3.2, you reported the IC50 to be 48.95 and 48.35 mg/ml for 4T1 and B16-F10 cells, respectively.

b) In the legend of Fig2, IC50 was reported to be 48.95 and 48.35 µg/ml for 4T1 and B16-F10 cells, respectively which is 1000 times more potent than the IC50 mentioned in section 3.2

c) In sections 2.5, 2.6 and 2.7, IC50 is reported as 34.1 and 36.4 µg/ml, for 4T1 and B16-F10, respectively which is different from both IC50s mentioned in section 3.2 and Fig2. 

3. What was the maximum tolerated dose for your extract? why did you choose IP route for administration? Why you used different doses for in vivo efficacy study against 4T1 and B16-F10 xenograft models?

4. Your extract produced similar in vitro anti-proliferative efficacy (IC50 in section 3.2), but failed to translate to similar response in vivo against B16-F10. Apart from the pharmacodynamic explanations that you provided, I would suggest to investigate if there are differences in pharmacokinetic parameters between the two tumour microenvironments. It would have been also useful if you had done some western blot to investigate differences at protein level (if any), particularly those involved in cell cycle, transcription and apoptosis regulation.

Author Response

We appreciate and welcome your review, as well as all of your comments and suggestions. Here is an answer to each of them:

Point 1. Is there use of the plant by traditional medicine practitioners or locals for treatment of cancer in your country?

Response 1: We are currently developing a research project in which we evaluate the antitumor effect of different plants widely distributed in our country. T. usneoides is abundant in our territory, and is traditionally used for colds and diabetes control. T. recurvata has been studied for its antitumor effect on different tumor lines and since they belong to the same family, we wanted to include it in the group of plants to be studied.

Point 2. How did you select 4T1 and B16-F10 cell lines for your study? did you test your extract against other cell lines? Given that your extract showed varying effects between the two cell lines, it would be logical to further investigate it against other cells, too.

Response 2: The cytotoxic activity of the ethanolic extract of T. usneoides was evaluated in the cell lines: MCF7 (22.08 ± 1.53), 4T1 (48.95 ± 3.74), B16-F10 (48.35 ± 5.53), K562 (17.16 ± 5.54), DA-3ER (14.2 ± 3.06), and U937 (120.85 ± 2.45). In order to evaluate their activity in vivo, the 4T1 and B16-F10 cell lines were chosen since we have these models actually standardized in our laboratory. Additionally, we are currently testing its activity in a murine model of acute myeloid leukemia with DA-3ER cells.

Point 3. I was confused and found it a bit annoying the way you reported IC50 values of your extract against 4T1 and B16-F10 cells, and it needs revision There is inconsistency, and hence, I would like to see the dose-response curve that you generated with Graphpad and the type of fit you used.

a) In section 3.2, you reported the IC50 to be 48.95 and 48.35 mg/ml for 4T1 and B16-F10 cells, respectively.

b) In the legend of Fig2, IC50 was reported to be 48.95 and 48.35 µg/ml for 4T1 and B16-F10 cells, respectively which is 1000 times more potent than the IC50 mentioned in section 3.2

c) In sections 2.5, 2.6 and 2.7, IC50 is reported as 34.1 and 36.4 µg/ml, for 4T1 and B16-F10, respectively which is different from both IC50s mentioned in section 3.2 and Fig2.

Response 3: In accordance with the suggestion, the dose-response curve was included as a supplementary figure (Supplementary Figure 1). Additionally, erroneous data was corrected, the correct IC50 is at µg/ml. Regarding point c), the IC50 that is being reported in these sections is the one used for the P2Et extract, hence the differences in the values.

Point 4. What was the maximum tolerated dose for your extract? why did you choose IP route for administration? Why you used different doses for in vivo efficacy study against 4T1 and B16-F10 xenograft models?

Response 4: The maximum tolerated dose for the extract was 2000 mg/kg for Balb/c and C57BL/6 animals. The administration doses were corrected, 142.5 mg/Kg is used in both models, it is possible that there was a transcription error. Regarding the route of administration, it was subcutaneous for the melanoma model and intraperitoneal for the breast cancer model, because of these treatment administration routes, previously standardized in our laboratory, allow a better bioavailability of the phytomedicine in the animal. The 4T1 and B16-F10 models are not xenograph models. In fact, 4T1 tumor, was obtained from a BALB/c mouse and an orthotopic transplantation allows the developing of the tumor. It is also the same for the melanoma model, isolated from a C57BL/6 mouse. For one of our more developed extracts we are actually tested the oral route with very good results after nanoencapsulation.

Point 5. Your extract produced similar in vitro anti-proliferative efficacy (IC50 in section 3.2), but failed to translate to similar response in vivo against B16-F10. Apart from the pharmacodynamic explanations that you provided, I would suggest to investigate if there are differences in pharmacokinetic parameters between the two tumour microenvironments. It would have been also useful if you had done some western blot to investigate differences at protein level (if any), particularly those involved in cell cycle, transcription and apoptosis regulation.

Response 5: We appreciate your suggestion and we agree with the need to further investigate this differential activity in the two models. One of the points that we wanted to highlight is the result of glucose uptake. In fact, in B16-F10 tumor cells, glucose uptake was increased after 12 hours of treatment with T.usneoides extract, compared with any changes in 4T1 tumor cells. It means that the metabolic plasticity of these tumor cells is different and, in fact, the way in which apoptosis occurs must be different, which could impact the type of immune response generated. In addition, although in vitro proliferation results showed a decrease in the proliferation of B16-F10 cells treated with the extract, it is much more prudent than the result with 4T1 cells. These results suggest that the differences observed in the in vivo models may be due to the metabolic differences of each tumor. Because of these differences, we are actually studding the fine mechanisms involved in cell death induced by different plan extracts.

Round 2

Reviewer 2 Report

Reviewer report-2

Author has been answered all the questions and added the content in the manuscript where it needed.

I am satisfied with author answers.

I would recommend this manuscript entitled " Tillandsia usneoides extract decreases the primary tumor in a 2 murine breast cancer model but not in melanoma" for publication. 

Please let me know if you have any questions.

Thank you.

Regards,

Prabhu Ramamoorthy